# An Edge Computing Application of Fundamental Frequency Extraction for Ocean Currents and Waves

**DOI:** 10.3390/s24051358

**Published:** 2024-02-20

**Authors:** Nieves G. Hernandez-Gonzalez, Juan Montiel-Caminos, Javier Sosa, Juan A. Montiel-Nelson

**Affiliations:** Institute for Applied Microelectronics (IUMA), University of Las Palmas de Gran Canaria, 35015 Las Palmas de Gran Canaria, Spain; nieves@iuma.ulpgc.es (N.G.H.-G.); juan.montiel@ulpgc.es (J.M.-C.); montiel@iuma.ulpgc.es (J.A.M.-N.)

**Keywords:** frequency parameters extraction, ocean tides and waves, underwater sensors, edge computing, offshore aquaculture infrastructures

## Abstract

This paper describes the design and optimization of a smart algorithm based on artificial intelligence to increase the accuracy of an ocean water current meter. The main purpose of water current meters is to obtain the fundamental frequency of the ocean waves and currents. The limiting factor in those underwater applications is power consumption and that is the reason to use only ultra-low power microcontrollers. On the other hand, nowadays extraction algorithms assume that the processed signal is defined in a fixed bandwidth. In our approach, belonging to the edge computing research area, we use a deep neural network to determine the narrow bandwidth for filtering the fundamental frequency of the ocean waves and currents on board instruments. The proposed solution is implemented on an 8 MHz ARM Cortex-M0+ microcontroller without a floating point unit requiring only 9.54 ms in the worst case based on a deep neural network solution. Compared to a greedy algorithm in terms of computational effort, our worst-case approach is 1.81 times faster than a fast Fourier transform with a length of 32 samples. The proposed solution is 2.33 times better when an artificial neural network approach is adopted.

## 1. Introduction

A current meter is an instrument used to measure the flow of water, in oceanography. Regardless of the sensing methodology used to obtain the measurement (e.g., mechanical, tilt, acoustic, or electromagnetic), the objective of the instrument is to obtain the velocity and direction of water flow [1].

Today, there are multiple applications in which this type of instruments is required. The measurement of flows in rivers or at the mouths of dams is a clear example of use [2]. Other very common use scenarios arise in the offshore industry, where it is required to measure the flows of water currents to which one or more moorings are subjected, whether on oil platforms, in aquaculture facilities or on a single buoy [3].

In any real-time monitoring application, as the number of involved instruments increases, the required bandwidth grows with the size of the sensor network. Moreover, a hostile environment for electronics such as water, and in particular salt water, is known to considerably reduce the main features of current electronic communications techniques. In terms of communications, wireless solutions are very limited by their attenuation over distance and most applications are related to low frequencies and close distances. Finally, submarine cable solutions significantly increase installation and maintenance repair costs [4].

Whatever type of approach is adopted, the optimal solution is to reduce the bandwidth requirements. In this sense, by the edge computing application in underwater sensor networks the raw data from sensors are processed on board for extracting those main characteristics which are demanded and, therefore, decreasing bandwidth requirements and communications power consumption [5].

On the other hand, in wave mechanics, the key features of a traveling wave are the fundamental frequency/harmonic, its amplitude and the velocity group. In general, those characteristics are obtained in desktop computers or cloud computing systems. There are some approaches in the literature to move, partially or totally, the extraction of those features on edge/on board [6]. However, the main drawback when applying edge computing is the scarcity of computing resources as well as ultra-low power consumption considerations.

Finally, the use of Artificial Intelligence (AI) to perform feature extraction in real-world applications has proven useful in time series data analysis applications. But, approaches using edge computing and artificial intelligence are very limited due to the required computing resources. There are some initiatives to facilitate the implementation of AI approaches on those kind of devices [7,8].

To put it briefly, the integration of edge computing in offshore aquaculture sensor networks offers numerous benefits. The application of AI in categorizing the gathered signals boosts the process of feature extraction.

However, currently there is no contribution in the literature that allows the designing and implementation of an AI-based algorithm for an ultra-low power microcontroller such as an ARM Cortex-M0+. In this paper, we study the design and implementation of an AI-based algorithm to select the adequate narrow pre-filtering at the input stage of an edge computing application to measure the ocean water current flow using the mentioned microcontroller platform.

Originally, the application is a sensor network where each sensor node identifies the fundamental frequency of the ocean water current on board by directly processing the acquired RAW data in real time. The main target of this research is to implement an embedded on-board algorithm to determine the narrow band pass of the acquired signal before its processing by the fundamental frequency extraction process.

This paper addresses the problem of designing an AI-based classifier algorithm on an ultra-low power microcontroller without a floating point or DSP hardware support. Furthermore, the efficiency of the proposed AI-based classifier plays a crucial role in preserving the original sampling time and edge computing of the sensor node/instrument.

The main contributions of this paper follows:An AI approach based on neural networks (NN) for narrow filtering of the input signal to a water current meter is proposed.The variation of the frequency of ocean waves are being scrutinized in depth.The implementation of complex neural network functionalities using integer, double and float type variables is studied in detail.

This paper is organized as follows: Section 2 introduces the previous works in NNs for embedded applications and parameters extraction from sensors signals. Some required background about the signal feature extraction is provided in Section 3. Then, experiments and discussion around the training process and the data distribution in terms of frequency variation are exposed in Section 4. It also includes the practical implementation, in an ARM Cortex-M0+ without a floating point unit, of our proposed AI approach. Comparisons between our implementations and a fast Fourier transform approach are also provided. Finally, the last section presents the main conclusions.

## 2. Related Works

In the literature, there are two methodologies to extract the fundamental frequency of a signal: one involves time or frequency domain-specific algorithms that provides the desired frequency value, while the other employs techniques based on curve fitting in time domain. The following paragraphs enumerates these contributions, focusing on their requirements in terms of computing resources and identifying which ones implement pre-filtering in their solution.

### 2.1. Signal-Parameter Extraction

There are multiple approaches in the literature to determining the fundamental frequency of a signal. Most solutions are based on the use of the Fast Fourier Transform (FFT) [9,10], Wavelet Transform [11], Hilbert Transform [12], least squares optimization, a Kalman filter [13,14] and/or Finite Impulse Response(FIR) filter [15]. Alternatively, the literature also presents related works that employ specific strategies. These include heuristic solutions like the Genetic Algorithm (GA) [16,17], Neural Networks (NN) [18,19] or zero crossing detector algorithms.

The most notable approaches in the literature are listed in Table 1. In general, the presented approaches are focused on obtaining a solution regardless of the computational effort involved. Proof of this is that the solutions adopted in the literature require the use of complex or real variables in their algorithms. This idea is also reinforced because the implementations of these algorithms are carried out in very high-level languages such as Matlab. Of course, there is some solution implemented in an embedded system [13], but it requires the use of a field programmable gate array (FPGA) to speed up some operations and also a floating point unit (FPU).

The clearest example in the literature of signal parameter extraction executed using edge computing is probably human wearable devices. Its restrictions in terms of size, weight, volume and mobility, as well as its low cost, define very restrictive design conditions for the solutions. The selection of a battery and a microcontroller becomes crucial as they pose conflicting design variables. Such is the case of battery capacity, power consumption, size and computing effort [23].

Traditional methodologies based on FFT or discrete wavelet transform (DWT) have been implemented using ARM microcontrollers. However, the target microcontroler is based on ARM Cortex-M4 which run at 168 MHz and includes a FPU. There exist other approaches like presented in [13] based on a Kalman filtering technique and the usage of an adaptive threshold algorithm for QRS complex detection. Despite its proven usefulness, the requirements in terms of computational effort remain high, compared to FFT-based techniques, due to the need to include several filtering stages before applying a peak detection algorithm and then feeding the Kalman filter.

From the point of view of the parameters extraction from a signal, the authors in [18] propose a solution based on Deep Neural Network (DNN) to obtain time-frequency features for the analysis and classification of electroencephalography (EEG) signals. Although classification is performed using a DNN, this approach focuses on high-level comparison of feature extraction algorithms. In a similar way, the authors of [19] present a Fault Diagnosis method based on a CNN to detect the variation in the signal features.

Regardless of the solution adopted to obtain the fundamental frequency, all approaches assume that it is mandatory to filter the signal before being processed. As was appointed previously, this input filtering has fixed its characteristic center/cutoff frequency and bandpass. Additionally, there are some approaches applied to fundamental frequency extraction in noisy conditions where the signal is preprocessed to determine the best filter selection from an available filter bank [21,22].

### 2.2. Embedded Time Series Applications

On the other hand, determining the fundamental frequency of a signal is basically a time series application. Regardless of the target device (e.g., cloud computing, GPU, CPU or microcontroller), data processing in a time series approach is performed in time or frequency domains. However, despite the proven usefulness of frequency-based solutions, since most of the sensor provides time domain data, the first step to apply this type of approaches involves the use of the Fast Fourier Transform (FFT). By definition, the FFT is a computationally intensive operation, and that is the main reason to avoid its use [24].

Furthermore, we must keep in mind that both industrial solutions and edge computing research use a wide spectrum of target devices in their solutions. These range from an ultra-low-power ARM Cortex-M0-based microcontroller to powerful GPUs or dedicated hardware such as a Field Programmable Gate Array (FPGA) [25].

Even though a GPU-based target device has its own restrictions, they are much less than ultra-low-power microcontrollers in terms of memory, computational effort and power consumption. For example, this is the case of the Jetson platform which includes a Pascal-based NVIDIA streaming multiprocessor with 128 or 256 CUDA cores and a Quad-Core Arm A57 and up to 8 GB of RAM [26]. In this type of cutting-edge computing device, the development and implementation of an AI solution is quite straightforward from, for example, widely used tools such as Matlab or Tensor Flow [27].

Approaches in the literature that are implemented on microcontrollers rely on CNNs, RNNs, LSTM and their variations [24]. These applications are related, for example, to the estimation of Ion-Li battery parameters [5], structural health monitoring [28], rail vehicle running states [29], ECG Monitoring [30] or the eye blink detection [31]. All of those approaches use at least one ARM Cortex-M4 that includes FPU and specific DSP instructions.

In addition to ad hoc solutions in which researchers or the industry present a specific solution for a particular application, there are multiple approaches to defining a framework of design tools ready to translate from a high-level description of an application to a hardware implementation.

An example of that is the Cortex Microcontroller Software Interface Standard Neural Network (CMSIS-NN) software library version 5.0.0. The main goal of this library is to minimize the memory footprint. Developed by ARM, this open source approach provides several processing primitives to implement neural networks in Cortex-M processors. In addition to the considerable support provided by a microcontroller company, its input interface is for learning software such as Tensorflow v2.15 and the Keras v3.0 wrapper to define the neural networks or commercial software like MatLab R2023b. The implementation is optimized for each series of Cortex-M. The ultra-low power Cortex M0 series requires a fully software-based solution. This is one of the reasons why this type of processor is not used in the literature. But, if the Cortex-M includes Digital Signal Processor (DSP) extensions or Arm M-profile Vector Extension (MVE), the approach uses those instructions to speed up the execution. This is the case of the M4 and higher versions.

There are other initiatives such as providing a complete solution to provide the learning environment and the implementation of the neural network on a target hardware implementation. For example, Microsoft provides an Integrated Learning Library (ELL). This open-source solution provides support for AI and machine learning on single-board computers such as Raspberry Pi, Arduino and micro:bit. Other examples are TinyML [32], TensorFlow Lite (TFL) [33], STM32Cube.AI [34] or NanoEdge AI Studio [35].

They are generally defined as learning and development environments for neural networks to be implemented on an edge computing device or microcontroller. Those all-in-one approaches provide elegant solutions at high levels that can be tested on embedded systems. Unfortunately, the list of target systems is very limited and moreover all of them are oriented towards cloud computing or embedded systems without power consumption restrictions.

As a summary and review of the literature, only [6] proposes a fundamental frequency extraction algorithm designed and implemented on an ultra-low power platform such as an ARM Cortex-M0+. Other approaches require the use of complex mathematical functions such as Fourier, Hilbert and Wavelet transforms, mainly described at high levels using MatLab, without obtaining practical results in edge computing applications. The minimum microcontroller supported by the published approaches is an ARM Cortex-M4 that includes DSP and floating point data operations supported by specific hardware. Moreover, none of those solutions are intended working scenarios where the energy consumption is a restrictive design variable.

Finally, despite the advantages of narrow pre-filtering applied to noisy signals in fundamental frequency extraction, its implementation is complex in terms of the mathematical operations required and only a small set of approaches use it at a high levels in Matlab [11,21,22]. In this sense, this research proposes the design and implementation of an intelligent algorithm based on AI to determine the best narrow band of the incoming signal and thus facilitate the extraction of the fundamental frequency. The proposed target platform is an ultra-low power microcontroller based on ARM Cortex-M0+.

## 3. Materials and Methods

### 3.1. Background

There are three main methods for measuring ocean currents and waves: tilt, acoustic and propeller-based techniques. Each of them has their own measurement peculiarities. Probably the most used, due to is low cost and resolution, is the tilt-based instrument. This research uses the tilt-based device presented in [36], which includes the algorithm described in [6] on edge computing.

Ocean currents are informally described as a series of water movements that change randomly over time. However, the literature mathematically describes the ocean currents as a Fourier series [37].
(1)$$\eta \left(x,y,z,t\right)=Re\left\{\hat{\phi }\left(x,y,z\right)e^{j\omega_{m}t}\right\}$$

In this formulation, $\hat{\phi }\left(x,y,z\right)$ models the spatial dependence of the velocity amplitude. It is remarkable that it is a three dimensional vector function. On the other hand, the exponential function describes its dependence on frequency. This generalized model can be applied to all types of ocean waves.

However, the real behavior of the ocean defines several scenarios that simplify this general model. For example, it is well known that the speed of water at the ocean surface is mainly due to wind. Using fluid mechanics, it is possible to determine the maximum height of a wave in terms of its length. Exceeding this limit implies the collapse of the wave ($h_{\omega }\left(collapse\right)\geq l_{\omega }/7$) [38]. Another obvious limit is defined by the ocean floor. The velocity of water flow at the seabed is zero. Therefore, the water flow velocity reduces its value as the depth increases.

On the other hand, assuming that the surface movements of water are produced mainly by the wind, it is well known that as depth increases, the movement is caused by the tides, which is the attraction of the moon and the sun. These waters that are not affected by the wind are called deep waters. The main characteristic that differentiates them from the rest is that the vector $\hat{\phi }$ becomes parallel to the surface of the ocean floor. The literature defines this limit as half of the length of the wave ($d_{\omega }=l_{w}\left(z=0\right)/2$).

Finally, since the attraction of the moon and sun is always present and the earth is rotating, ocean water is always in motion.

### 3.2. Feature Extraction Procedure

Given a deep-water location, Equation (Equation 1) is defined as follows:(2)$$|\hat{u}\left(t\right)|=a_{0}+\sum_{i=1}^{K}\left[a_{i}sen\left(2\pi ift\right)+b_{i}cos\left(2\pi ift\right)\right]$$

This equation describes the magnitude behavior of the water velocity in a desired location. The constant coefficient $a_{0}$ models the movement of the water flow when there are no oscillations. Summation is basically the Fourier series formulation of water flow variations based on harmonic motions. *f* is called fundamental frequency. It represents the minimum frequency component that makes up the Fourier series that model the wave. Higher frequencies are multiples of the fundamental frequency coefficients. $a_{i}$ and $b_{i}$ are the amplitude of the involved frequency components [39].

According to related theory, the number of sinusoidal components of the Fourier series *K* is infinite. However, in practical scenarios this number is finite because the amplitudes of the high–frequency components are negligible. On the other hand, $a_{0}$ is quite simple to obtain. It is basically calculated as the average over a period of time of the acquired signal.

The coefficients $a_{i}$, $b_{i}$, *f* and *K* are the targets of the feature extraction procedure.

Figure 1a shows the classic feature extraction execution scheme [6,9,10,12,13,14,15,16,17,20]. It is a straightforward procedure. The first stage is acquiring the raw data from the sensor system. Depending on the capabilities of the acquisition system, this first stage sometimes includes filtering, normally called pre-filtering, e.g., low-pass or band-pass filters, among others. The second stage is itself a specific filtering stage. It is mandatory because the pre-filtering performed in the sensor acquisition system generally does not fit the requirements of the final application. The last stage is the feature extraction algorithm.

The ultra-low power implementation of the propose signal feature extraction scheme with an embedded system is focused on reducing the computational effort for all the stages. The first idea is to reuse the pre-filtering stage of the acquisition system as much as possible. In general, those pre-filtering stages are more energy consumption efficient than equivalent software filtering on microcontroller units. Therefore, this computing resource is not negligible.

As a disadvantage, the use of filtering at this early stage entails the irreparable loss of the original raw data, having only access to the already filtered data. Moreover, extracting features from a signal in many applications requires both filtered and raw data. For example, processing of filtered data provides the points of interest in time and/or frequency domains, and then the amplitude of those locations are obtained from the raw data. In this sense, the pre–filtering at the acquisition stage is mainly used to eliminate unwanted signals and/or noise from adjacent bands to the one of interest.

As is shown in Figure 1b, a feature extractor is designed to perform the search for a target feature within a defined bandwidth. Depending on the search range and the desired precision, the proposed procedure searches the fundamental frequencies on certain bands (subbands) of the complete search space. As the search space grows, there are more subbands, and any signal within the search space that does not belong to the subband of interest acts as noise from the extraction algorithm point of view [11,21,22].

### 3.3. Involved Frequencies

At this point, it is necessary to know the range of frequencies which are expected by the sensor system. Since the application is to measure the water flow in deep water, the expected frequency range is directly imposed by the range of velocities to be measured. As noted in previous sections, the minimum speed of water is at the bottom of the sea and is zero. This theoretical lower limit is exceeded by the resolution and/or noise floor of the instrument. On the other hand, although there are water meters capable of measuring up to 7.0 m/s in ocean surface applications, the same equipment in deep waters does not measure more than 1 m/s. Note that this value represents the magnitude of the velocity to be measured.

The frequency at which the velocity of the water flow varies depends directly on the length and height of the wave. The equations that model this behavior are the following:(3)$$\lambda =\frac{2\pi }{k}=\frac{2\pi g}{\omega^{2}},$$
and
(4)$$\omega =\frac{g}{v_{p}}tanh\left(kh\right).$$
where ω is the frequency in radians (2πf), *g* is the gravity vector and *k* is the wave number, called angular repetency. In Equation (Equation 4), $v_{p}$ is the velocity phase of the wave and *h* is the wave height.

At deep water, kh≫1 and therefore tanh(kh)≈1, then Equation (Equation 4) is rewritten as
(5)$$f_{p}=\frac{g}{2\pi v_{p}}.$$

One fact to take into account based on Equation (Equation 5) is that the frequency of water flow variations does not depend on depth. On the other hand, if the water flow is composed of dispersive waves, we need to define the group velocity. It is defined at deep water as half of the phase velocity ($v_{g}=\frac{v_{p}}{2}$). Based on that, for a dispersive wave at velocity of 1 m/s the range of frequencies goes from 1.56 Hz down to 0.78 Hz.

### 3.4. Filtering

Figure 2 shows the response of an FIR filter using a power of two as the length. This is the typical response of mean filtering which is performed in sensor acquisition systems. The sampling frequency of our application is 12.5 Hz. Therefore, following the Nyquist criterion, the maximum frequency of a correctly acquired sinusoidal signal is 6.25 Hz.

In this paper, we use low–pass FIR filtering to build our filter bank. The reason for this is its low requirement in terms of computational effort compared to other digital filter topologies. In our case, we choose the filtering as follows [6]:(6)$$FIR_{L}=\frac{1}{2^{L}}\sum_{i=0}^{2^{L}-1}x\left[n-i\right]-\frac{1}{2^{\left(L+1\right)}}\sum_{i=0}^{2^{\left(L+1\right)}-1}x\left[n-i\right]$$

Figure 3 depicts the attenuation response of the set of filters following Equation (Equation 6), when *L* is set from 2 to 11. This filter bank is made up of 10 FIR filters. Each filter is labeled with a letter from A to J, represented from right to left in Figure 3, respectively. Note that despite the fact that the filter is identified with a length *L*, the required number of samples is the double, as shown in Equation (Equation 6).

Figure 3 also illustrates the half power level, which is the −3 dB mark. It is remarkable that using the half-power level as filter limit, each filter overlaps some band of the other filters. The cutoff frequencies at −3 dB of each FIR filter are enumerated on Table 2. However, in this research we have chosen this set of filters because it is possible to define the filter bank without overlap if we select the correct attenuation level −0.58 dB, which represents an absolute gain of 0.875, a loss of only 12.5% in comparison to a 50% for −3 dB, in the worst case.

### 3.5. AI-Based Solutions

Probably of all the possible AI-based solutions, a neural network approach to calculation or classification is the most widely used in industry and research [24]. In the sensors research area, its use to process a time series of data is probably one of the most used. Data time series refer to data which are acquired at successive sampling times. Therefore, the purpose of time series processing is to obtain certain parameters that depend on the measurement and its evolution over time.

The computational complexity to extract the desired parameters from the time series data is achieved by employing different hidden layers in the neural network implementation. In this paper, regardless of the target application, we must distinguish between the learning process and the neural network. From our point of view, a neural network is just an algorithm based on mathematical formulas. The learning process is an algorithm that obtain the coefficients, weights and biases of the mathematical formulas defined in the neural network. Although there are multiple approaches to integrate both algorithms into a single solution, the resources in terms of memory and computational requirements are prohibitive when considering implementing such solutions into ultra-low power microcontrollers.

In the literature, there are five main types of neural network architectures which are applied to time series problems; they are Multi-Layer Perceptron (MLP), Convolutional Neural Networks (CNNs), Recurrent Neural Networks (RNNs), Long Short-Term Memory (LSTM) and Graph Neural Networks (GNN). Of course, we can consider that there is a sixth class of neural networks that includes all approaches that hybridize two or more of the other classes [40].

The MLP defines quite simple architecture. The basic one is made up of 3 layers, called input, hidden and output layers. In each layer, all neurons are connected to all neurons in the previous and next layers. If the neural network includes more than one hidden layer, the MLP is called an artificial neural network (ANN).

A CNN is the evolution of the MLP/ANN. Its architecture is based on including a sequence of mathematical operations between layers of neurons to filter the processed data. Those operations involve convolution, pooling and activation functions. From a mathematical point of view, a CNN includes non-linear functionalities such as statistical functions, e.g., average, standard deviation or advanced mathematical functions like exponential and square root functions. In terms of computational complexity, since MLP is basically based on linear transformation functions, the use of CNN architecture greatly increases the complexity of implementing a solution on an ultra-low-power microcontroller.

A RNN is based on a CNN and including some feedback for remembering information computed previously. This input time-dependent solution has demonstrated usefulness for sequence prediction problems. However, they suffer the so-called exploding and vanishing gradient problems. To process the same length of input data, a RNN requires similar computational effort as CNN-based solutions, although it requires more memory resources.

LSTM is basically an upgrade of the RNN architecture. A RNN defines a computing unit that includes the neuron, an input gate, an output gate and a forget gate. This last function helps the neural network to promote a set of inputs against the influence of other inputs. The main problem with the RNN architecture is that all processed data in the time series have the same importance. However, it is well known that some applications require modulating the contribution to the neural network evaluation in some periods of the time series.

Finally, a GNN defines a graph-based architecture, where the layer structure is not clearly present. Its network structure has the same characteristics as other graphs in the literature (e.g., cyclic or not, directed, undirected or any combination). Since there is no predetermined internal structure, to obtain an implementation of the NN structure is considered as an input to the learning process. The learning process requires great attention to achieve the tuning of the weights and biases.

As a summary, it is possible to apply any neural network architecture to process a time series and extract a specific characteristic from a signal. However, of all those are discussed, and due to their reduced complexity in terms of training or implementation, DNNs are the most used.

### 3.6. Deep Neural Network

The key point of this work is to design and implement a neural network to choose the correct filter from a filter bank. The pre–processing of the incoming signal is the input data of the neural network. The output is the selected filter. In this sense, Figure 4 presents the proposed DNN scheme used for the water current meter. The input layer receives N contiguous samples from the sensor system. These acquired data are then processed using a full connected convolution layer (see Convolution 1 in Figure 4b). An activation layer then shifts the processed data for the next stage. The fourth layer normalizes the results from the previous activation layer.

The second to fourth layers confirm the first feature extraction stage (FE Stage 1). After that, we included a second feature extraction stage with identical topology. That is, a full connected layer, its activation layer and finally a normalization layer. Both feature extraction (FE) and classification phases are introduced in the proposed DNN implementation. A pooling stage is in between the FE and CLASS phases

The classification is conformed by a full connected layer, its activation layer and, finally, the output layer.

## 4. Experiments and Discussion

### 4.1. Training Data

Probably one of the most important issues when creating a neural network application is the availability of data and their representativeness. In our case, the target application is an industrial solution called 24/7. That is, the application is designed to work in real time all the time. The diversity of scenarios in an offshore aquaculture infrastructure is high. However, the available ocean data at this location are limited to only 180 days, from 5 May to 1 November. The dates to carry out the measurement campaign were determined based on the know-how of Acuanaria S.L, owner of the aquaculture facilities.

The raw data used in this research were acquired over a period of 180 days starting on 5 May 2021 at 12:00 noon using the deep water current meter described in [36]. This instrument was deployed at 15 m depth at coordinates 28°3′46″ S and 22′46.48″ W in an offshore area close to Gran Canaria in Canary Islands (Spain). The location of the aquaculture infrastructure, where we deployed the instrument, is located in the North Atlantic Ocean and near the west of the African continent. This location is mainly dominated by the global circulation flow of ocean waters called the Canary Current. This water flow follows a north–south direction pattern.

Figure 5 presents one day of frequency measurement in periods of 2.56 s. The blue bars show the density distribution of the highest frequencies found and the orange bars show the distribution of the extracted fundamental frequencies. Note that the main goal of this investigation is to determine the fundamental frequency of a deep water current flow. Therefore, the objective of this work is to determine the frequencies in the pass band defined by the orange curve. On the other hand, the blue function appears because the measured waves have dispersive behavior and, therefore, multiple components of the fundamental frequency arise (see the factor if in Equation (Equation 2) for more details).

Although the sampling rate is 12.5 Hz and the maximum detectable frequency with the Nyquist criteria is 6.25 Hz, we note that from a practical point of view, frequency components above 2.5 Hz are negligible. The distribution function of the maximum components is constituted by a wide spread band pass curve with three more or less equal peaks. However, the distribution of fundamental frequencies follows a concentrated band-pass curve with a single central frequency.

Although both bands are clearly differentiable, it must be remembered that only the fundamental and maximum frequencies are represented. According to Equation (Equation 2), there are other frequency components among those shown. Figure 6 presents the distribution function of the number of components obtained for the same data presented in Figure 5.

An interesting conclusion from Figure 6 is that 93.88% of the measured data is composed by at least 3 and no more than 6 frequency components. Only a 0.2% of the measured waves in one day are alone waves, the other 99.8% are dispersive waves. These curves are quite similar in all the measured data which were processed during the field campaign. The maximum difference between what is presented in previous figures and the rest of the processed values is always below 15%.

### 4.2. Application Interface

#### 4.2.1. Output

Based on the histogram shown in Figure 6 and the bank filter presented in Figure 3, the selection is made between 4 filters. Figure 7 presents the density function of the fundamental frequencies introduced, previously, with a resolution/bin of 0.1 Hz. Also, this figure includes as vertical black lines the limits of the set of filters from the bank of filters. The involved filters are labeled from C up to F. Their length in terms of samples are from 16 up to 128 (see Table 2).

Based on that, the number of outputs of our DNN must be 4 classes. However, from a purely empirical point of view, we conclude that the fundamental frequency is located in most measurements carried out between 0.2 and 0.4 Hz. However, in that bandwidth we only have a single filter bank available. To improve the fundamental frequency feature extraction procedure, we divide this central band, where the target variable is most likely to be found, into 4 virtual bandwidths. The proposed additional bandwidths are outlined in Figure 7 with vertical dotted lines. Therefore, the proposed DNN application has 7 classes based on this filtering criterion.

#### 4.2.2. Inputs

Once the number of outputs of the DNN has been specified, the next key question is to define its number of inputs. However, before continuing, it is necessary to clarify how the frequencies of a signal is determined from samples of it over time because depending on the technique, it severely impacts the requirements when determining the amount of data to be acquired.

There exist three possible mathematical techniques to determine the fundamental frequency of a signal. They are Fast Fourier Transform (FFT), model fitting (MF) and peak detector (PD). PD techniques are useful for signals with very repetitive patterns and in most cases with a high signal-to-noise ratio. On the other hand, MF methodologies are highly model dependent. This application is a clear example where an MF technique can be applied. The FFT algorithm is an elegant mathematical way to obtain the frequency components of a signal. However, the computational effort is a problem when the target application must run on an embedded system, in particular, on an ultra-low power microcontroller without FPU. Finally, PD algorithms are not easy to apply for ocean water flow measurement due to their non-repeatable characteristic.

From a practical point of view, a large number of samples to process implies a long measurement period, and therefore, the greater the probability that the fundamental frequency will change. On the other hand, a low number of samples reduces the frequency resolution when for example a Fast Fourier Transform (FFT) is used to determine the fundamental frequency.

The used instrument has a sampling frequency of 12.5 Hz and it is not configurable. The resolution in frequency Δf for FFT methodology is $f_{sample}$/L. For example, a sample length of 128 implies a resolution of 0.098 Hz, that is, practically 0.1 Hz. But probably, the key problem when using the FFT with a short sampling period is the windowing problem. Because the original signal is truncated, the FFT algorithm provides erroneous high-frequency components. Therefore, short periods favor this type of malfunction.

The resolution of MF methodologies is basically related to the approximation methodology used and the error supported, being in most cases much higher than the FFT for the same number of samples. As mentioned before, the probability of fundamental frequency change increases with time, the MF methodology requires more and more components as this case occurs.

Figure 8 illustrates the number of components required by a MF methodology for three different sample lengths, 16, 32, and 64 samples. They correspond to 1.28 s, 2.56 s, and 5.12 s, respectively. We observe that 1.28 s requires no more than 5 components and the average is close to 2.5 components. The 32 samples requires no more than 7 components, and 64 samples requires in most of the cases more than 8 components.

From the fundamental frequency extraction point of view, more than 8 components in the model indicate a high probability of change. For this reason we choose 32 or 16 samples. In addition, the modeling error between 16 and 32 are quite similar, bellow 10% on average. But in this work, in order to define the worst corner case we select 32, because it is more demanding in terms of computational effort than the 16 samples. Note that for a lenght of 16 samples, using an FFT implies a precision of 0.78 Hz. Therefore, the number of inputs of the proposed DNN in this work is 32.

### 4.3. Training Process

In order to train the architecture, we first build a database from the field campaign measurements. We used Matlab R2023b from Matworks to preprocess a total of 180 days of raw data. Due to the large amount of available data, the database is organized day by day. One day of raw data contains 1.08 M samples. Each sample includes at least the three-axis accelerations of the measuring device among other measured variables, e.g., gyroscope, magnetometer and temperature among others. The obtained database contains a total number of 194.4 M samples.

Since this research focuses on the extraction of the fundamental frequency parameter and not its amplitude, and because the frequency components of the measured water flow are present in the three measured orthogonal axes, this work only processes the accelerations of the vertical axis of the instrument. For this reason, we uses the Matlab fitting functionalities to obtain the fundamental frequencies and its multiple components over only one acceleration axis for a given sample length.

The raw data for a day are reorganized in time periods defined by the selected length. For each period, the fundamental frequency, other frequency components and their amplitudes are obtained according to Equation (Equation 2).

As was mentioned previously, the amount of data is huge in comparison with the number of classes/bandwidth to detect. On the other hand, the comparison of the available data reveals that although it is confirmed that the amplitudes of the water flow speed change substantially between time slots, days throughout the month and/or the complete measurement campaign of available data, the fundamental frequencies behave similarly between days. For this reason, we have trained the DNN with data from a single day.

A day has 1,080,000 samples at 12.5 Hz. When a length of 32 samples is set, the number of periods are 33,750. We used the 10% of the periods for validation and a 30% for testing purposes.

Figure 9 plots the accuracy and loss function of the training process for the proposed architecture when the number of fully connected neurons in Layers 1 and 2 is set to 16. In this approach, normalization Layers 1 and 2 perform a batch normalization function. After 2150 training iterations, the DNN converges to a accuracy of 100% with a lost about 0.5%. After this point, the optimization of the DNN continues reducing losses bellow 0.1%.

#### Optimization

This initial approach is based on using half of the number of inputs to build the full connected layers. The question is if this first approach can be optimized because the target instrument uses an ultra-low power microcontroller and in stand-alone working mode, the power source is a 3600 mAh Li-ion battery.

A first optimization is to reduce the number of used neurons. In this sense, we have made several trials with all possible combination of neurons from 16 down to 5 the FE Stages 1 and 2. Given a combination of multiple layers, each trial takes, approximately, 1.5 min and convergence is not guaranteed. We automate the training process to repeat each combination up to 50 times. The training for a given combination of number of layers stops if the trial converge. Our goal is not to obtain the best convergence, but also to determine if a solution exists.

As expected, as each layer has a smaller number of neurons, convergence takes longer. We obtained a DNN with 5 and 10 neurons in FE Stages 1 and 2. In our experiments, the second layer cannot be reduced more. On the other hand, the first layer can be reduced by up to 4 neurons, but the lost function arise up to 1% and the number of iterations must be increased to 10 k.

A second optimization step is to reduce the number of other non-neuron functionalities. In the proposed scheme (see Figure 4), the most expensive function in terms of computational effort are the normalization functions. However, normalization functions are critical to keep the intermediate processed data/features within a known range. Moreover, the literature pointed out that they also increase the speed of the convergence in the training process.

Our goal is not to define a new normalization function, but one open question arises whether its use is necessary. In this sense, we have studied the computing process through each layer of the DNN. First of all, the nature of the measurement process is analyzed. The acquired data are obtained from the vertical acceleration axis of a water current meter located in an ocean offshore infrastructure. Due to the gravitational tide, the water flow never becomes zero. From the point of view of the acquired data, this fact implies that the sign of the data is never swapped. In other words, the water flow meter always provides a positive or negative value, but never changes sign.

The first layer, labeled Full Connected 1 in Figure 4, is composed by linear functions. We observed in all obtained solutions that layer weights has the same sign. The difference between those weights are close to a decade in the range of [−0.43, 0.53] and the average is 0.02. On the other hand, Figure 10 shows the input distribution. It follows a normal distribution. Based on these characteristic, since the normalization objective is to conform a normal distribution, the input data follows a normal distribution and the processing is performed using linear function. Therefore, we propose to remove Normalization 1 layer.

The training of the modified DNN converges in the same way without the removed layer. On the other hand, after verifying the solution we observe that one of the weights of the Full Connected 1 layer in this solution has an abnormal variation. Most of the values are in the same range and there is one value that is 4 decades lower. According to our knowledge, since the range of input values is the same in all inputs, when this behavior appears it means that there are more neurons than necessary.

Based on those clues, we remove Feature Extraction Stage 1 and run the training process again. This modification converts our proposed DNN into an artificial neural network (ANN). Therefore, from a practical point of view, this third proposal greatly reduces the computational effort. The training of the ANN is carried out using from 8 to 3 neurons in the Fully Connected 2 layer. Figure 11 plots the results for 3 neurons in the Fully Connected 2 layer.

Despite the reduction in the number of neurons, in the Fully Connected 2 layer, the training process converges using only 3 neurons. However, this convergence is achieved one time every 9 or more training trials. As expected, reducing the number of neurons pushes the training process to the limit. To have a better chance of success, we increased the total number of iterations from 5000 to 10,000. In Figure 11, we observe one example of the convergence. In comparison with the DNN training process, regardless of whether the validation process reaches the 100% percentage in the same way, the training process suffers convergence throughout the process. The proof of this is how the blue line in Figure 11 does not follow the smooth behavior.

The Loss function in Figure 11 reaches the same level than the DNN training. However, its convergence trajectory presents several peaks at those points where the training encountered difficulties in optimizing the neural network. It is remarkable the disturbance around iteration 4200 where the convergence of the complete network was reduced up to the 40%. Fortunately, the convergence reaches 10% constantly and the loss function is under 0.2 after iteration 6700.

### 4.4. Practical Implementation

The target of the implementation is an ultra-low power microcontroller from NXP. In particular, the device is the MKL17z256. This microcontroller incorporates a ARM Cortex M0+ without floating point unit (FPU). Its memory is 256 KB and 32 KB of Flash and SRAM, respectively. The reason to select this device is because it was used in [6]. After designing at high level the DNN architecture and testing its usefulness, next step is to study in detail the translation from high level to the low level of the target system.

#### 4.4.1. Input Layer

As mentioned before, the input to the proposed DNN scheme is directly the data acquired from the sensor system. However, we must take into account not only the nature of the data processed, but also the data domains handled by the acquisition system. In other words, from a high-level point of view, this measurement application belongs to the domain of real numbers. It involves the use of floating-point representations, such as double or floating data types. However, the digital nature of the acquisition system results in direct translation from the domain of real numbers to that of integers. This implies that the input layer can be fed directly with the sampled data as an array of integer values.

#### 4.4.2. Fully Connected Layer

This feed-forward layer, with *M* neurons and *L* input sources, follows the equation:(7)$$y_{j}\left(x\right)=\sum_{i=1}^{L}w_{ji}x_{i}+w_{j0};\forall j\in \left[1,M\right]$$

Each neuron of the layer is computed as the summation of each previous stage output/input source, weighted and biased with its own specific values. Each value from the previous stage in the integers range. The weight and bias values obtained from the neural network training procedure are in the range of real numbers.

Taking into consideration the range of the incoming data (see variable *x* in Equation (Equation 7)), it is defined at first by the digital output resolution of the acquisition system. However, we assume that this range is defined for a critical scenario that will never occur. An example of this is assuming the maximum expected measurement values are within the range of ±1.5 g, and the best full range of the acquisition system is ±2 g. If the resolution is 14 bits, the real range of the acquired data is approximately 13.58 bits.

The ARM Cortex M0+ includes an integer multiplication unit capable of executing this mathematical operation in a single cycle. But the multiplication unit returns the 32 Least Significant Bits (LSB) of the 64-bit result of the operation (32 bits × 32 bits).

In the worst case, we can assume that the width of the sampled data is equal to the maximum resolution, i.e., 14 bits. On the other hand, as was appointed previously, the weights and bias values are real numbers. Since our sensor system is in worst corner case within the 14 bits wide, we can warranty its non-overflow operation from the data point of view. On the other hand, in order to speed up the learning process we defined as real numbers the weight and bias values in our proposal.

But our implemented proposal is based on the use of integer values of those parameters. In this sense, we discretize the real values to a range of 16 bits according to the following equation:(8)$$w_{ji}=W_{ji}/2^{d_{ji}}+\xi_{ji},$$
where $W_{ji}$ is an integer obtained as the quotient of dividing the original weight or bias by a constant power of two ($2^{d_{ji}}$). $\xi_{ji}$ is the remainder of the division.

Assuming $\xi_{ji}$ as discretization error of $w_{ji}$, Equation (Equation 7) can be rewritten as follows:(9)$$y_{j}\left(x\right)=\sum_{i=1}^{L}\frac{W_{ji}xi}{2^{d_{ji}}}+\frac{W_{j0}}{2^{d_{j0}}};\forall j\in \left[1,M\right]$$

Since $W_{ji}$ and $2^{d_{ji}}$ are calculated in a desktop computer offline the real application, the implementation complexity of this Equation (Equation 9) is reduced considerably. Since all weights and bias are now defined in the integers domain, the multiplication can be performed in the microcontroller multiplication unit directly. On the other hand, division by the constant power of two ($2_{ji}^{d}$) can be easily implemented with an arithmetic right shift function. Of course, assuming the error due to the quantization.

#### 4.4.3. Activation Layer

From all possible activation layer, we selected by its implementation simplicity a Rectified Linear Unit (ReLu)). This layer performs a non-linear function as follows:(10)$$ReLu\left(x\right)=\;\{\begin{array}{cc} x, & x\geq 0\\ 0, & x<0 \end{array}$$

From the point of view of its translation to the microcontroller, this activation function is basically an if-conditional operation.

#### 4.4.4. Normalization Layer

The goal of this layer is to adapt the distribution of the output results of the previous layer to a distribution with a well-known mean and variance. In general, these statistical parameters are zero and one, respectively. From the point of view of computational effort, this layer is probably one of the most critical. Nowadays, normalization procedures involve the use of nonlinear functions such as exponential, square root, and division mathematical functions. The normalization function most used in the literature is Batch.

The goal of Batch normalization process is to adjust the average of the incoming data distribution to zero. Additionally, it encourages the standard deviation to approach one. Following equations defines this normalization:(11)$$\mu =\frac{1}{N}\sum_{j=1}^{N}x_{j}$$
(12)$$\sigma^{2}=\frac{1}{N}\sum_{j=1}^{N}{\left(x_{j}-\mu \right)}^{2}$$
(13)$$BatchNorm\left(x_{j}\right)=\gamma \left(\frac{x_{j}-\mu }{\sqrt{\sigma^{2}+\epsilon }}\right)+\beta$$

As normalization function, the number of inputs to process (*N*) is equal to the number of outputs to provide. Equations (Equation 11) and (Equation 12) compute the average and the standard deviation of the data to process. The *BatchNorm* function operates each input $x_{j}$ Equation (Equation 13). Note that this last equation includes a linear translation using scale and bias coefficients γ and β, respectively.

The average and standard deviation is computed over all training data instead of over the incoming data. In other words, Equation (Equation 13) can be rewritten as:(14)$$\lambda =\frac{\gamma }{\sqrt{\sigma^{2}+\epsilon }}$$
(15)$$\tau =\beta -\frac{\mu }{\sqrt{\sigma^{2}+\epsilon }}$$
(16)$$BatchNorm\left(x_{j}\right)=\lambda x_{j}+\tau$$

Where, λ and τ are calculated during the training process. Therefore, its implementation on a microcontroller only depends on the type of data used to represent the variables of our application in the same sense as previously explained in Section 4.4.2. There are other normalization functions, but they are all more computationally greedy since they are born from the function explained in the previous paragraphs.

#### 4.4.5. Activation Function

The most used activation function is the one called SoftMax. It is defined as follows:(17)$$SoftMax\left(x_{j}\right)=\frac{e^{x_{j}}}{\sum_{i=1}^{N}e^{x_{i}}}$$

SoftMax function is computed for each output of the previous stage ($x_{j}$). But, the denominator of Equation (Equation 17) is the same for all these calculations. A possible pseudocode implementation to reduce the computation is shown in Figure 12.

It is clear composed of two steps. Basically, the intuitive main loop to repeat the SoftMax function for each output is divided to precompute the exponential function of the numerator of each SoftMax evaluation and accumulate this computation to second loop. In second loop, the division of the SoftMax normalization is computed.

#### 4.4.6. Complex Math Functions

The execution of these kind of mathematical functions is widely studied in the literature. For example, the authors of [41] use a STM32F746 and STM32H750 from ST Microcontrollers including a FPU (float and double precision, respectively) to compare the implementation of an exponential function using hardware acceleration and the software solution, in terms of computational effort required. Although this study is evaluated using an Arm Cortex-M7, the authors conclude that there are no significant differences in the number of cycles required by the operation, regardless of bit width using integers or floating point numbers. The reason is that those mathematical functions are implemented using well-known algorithms and their are highly optimized for each data type.

Table 3 and Table 4 present the computational effort required to execute an exponential, square-root and division function. The data show in those two tables have been obtained on selected MKL17Z256 with an Cortex-M0+ without FPU from NXP Semiconductors. The difference between both tables is just the optimization flag in compilation time.

In the same way as presented for Arm Cortex-M7 in [41], the number of cycles to execute in an exponential or square root function is quite similar for the type of data evaluated. In case of the exponential function, the advantage of to use optimization flag level 2 is a minimum of a 26.8%. The square-root function requires a minimum of 24.8% less execution cycles using the level 2 optimization flag. Finally, the division requires a 30% less computational effort in same conditions than other math functions.

### 4.5. Computational Effort Comparison

Table 5 presents the computational effort of proposed DNN approach when it is implemented using a ARM Cortex-M0+. The evaluation have been performed over all captured raw data. That it is equivalent to more than 6 M evaluations.

The first extraction feature layer have been implemented using integer variables. However, the rest of the DNN approach is performed using float and double precision variables. Of course, these types of variables are supported by the GCC compiler and, since the selected microcontroller does not have an FPU, its implementation is based solely on software. In this Table 5, the last two columns present the machine cycles required for Float-Integer variable operations and the middle two columns the equivalent for Double-Integer.

Firstly, from the comparison between the required computational effort measured in CPU cycles, we observe that the operation with double type variables requires three times more cycles than when using integer variables. However, using float type variables only requires two times CPU cycles than same operation using integer type. Taking into account that a variable of a float type uses the same 32-bit memory width as the integer variable, the difference is basically due to the software implementation of the mathematical operations.

This difference is more significant as the complexity of the mathematical operations required increases. This is the case for example of the normalization function. Our proposal uses as Normalization layers 1 and 2 are a linear function (see Equation (Equation 16)) but use double and float-type variables. Given the same number of iterations and functions, the double operation requires more than twice as many resources as the float operation. Furthermore, the float operation requires 27.5% more CPU cycles compared to the entire operation.

The time required to execute the DNN on the ARM Cortex-M0+ at 8.0 MHz is 9.54 ms in case of using double type variables and 7.96 ms when a floating-type solution is adopted. It represents a 17% reduction. In terms of memory, the DNN requires twice memory pace for the double type approach compared to the float-type solution.

Table 6 shows the computational effort of our proposed ANN solution on an ARM Cortex-M0+. Note that we eliminated one of the two feature extraction steps. Furthermore, the initial feature extraction is performed using only three neurons instead of five as in our DNN approach. The last stages of polling and classification are performed identically in terms of functionality compared to the DNN solution. That is the reason why they do not change in terms of computational effort.

The ANN solution requires 8.51 ms and 7.41 ms for the operation of double-integer and float-integer variables, respectively. The advantage of using the double solution over the float one is only 13%, but the advantage of using the ANN-Integer-Float solution over DNN-Integer-Double is a 22.3%.

Demonstrated that it is possible to implement, on an ARM Cortex-M0+, a neural network to determine which filter best adapts to the input signal, we are left to compare it with some other methodology. As was appointed previously, there is not other approach in literature based on this ultra low power microcontroller. However, we can assume that because the number of samples in each time series is only 32, we can implement an integer FFT function and sort the results to determine the best filter to preprocess the input signal.

Finally, we have implemented the FFT using CMSIS–DSP library for ARM Cortex-M0+. We activated the optimization level 1 in compilation time using gcc. The 32 length FFT requites 17.29 ms in same conditions than our proposed approach. It represents 1.81 times more computational effort than our DNN-integer-double solution (worst case) and 2.33 times better when an ANN-integer-float approach is applied.

## 5. Conclusions

This article presented a unique artificial intelligence powered method for choosing the appropriate input filter from a filter bank. The intended application employed an extremely energy-efficient microcontroller, which was based on the ARM Cortex-M0+ but did not include a floating point unit. The research covered all the phases of the design of artificial intelligence from its early stages to its implementation in the instrument. The integration had been done without high-level design environments and quantization procedures.

Firstly, the use of a DNN designed to classify the signal from a MEMS accelerometer sensor within the bandwidths of an available filter bank was proposed. Each function involved and its parameters were studied in detail from a practical point of view. The ultra-low power implementation guided the design decisions up to obtain an optimal version of the DNN solution. From the study of the weights of the DNN approach, an ANN solution was proposed, which was also the object of study and optimization. The research conducted in this paper demonstrated the use of conventional DNN and ANN in ultra-low power devices in a real application.

In both proposed approaches, there were two main key points: the minimization of the number of neurons used and the type of variable used in the calculations. The training, validation and test processes had been carried out with real data from several marine water current sensors. The advantage of using the ANN solution over a traditional fast Fourier transform was 2.33 times better and only requires 7.41 ms to perform classification.

## Figures and Tables

**Figure 1 sensors-24-01358-f001:**
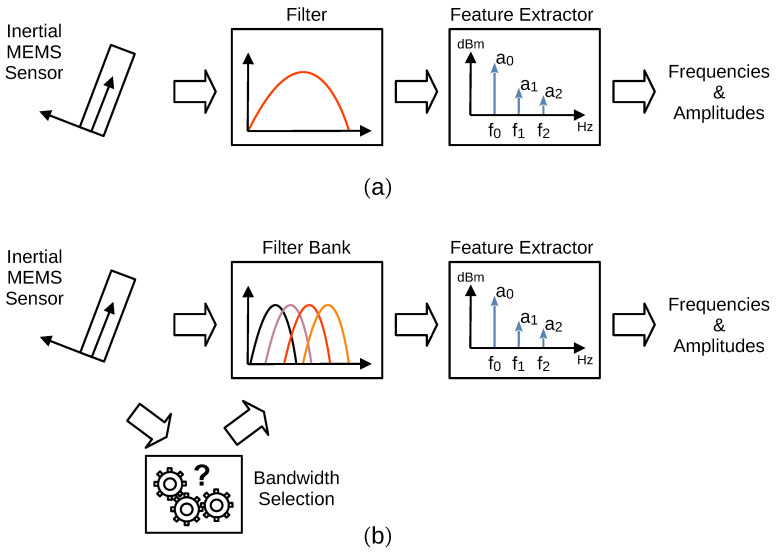
Signal feature extraction scheme, (**a**) traditional, and (**b**) the proposal.

**Figure 2 sensors-24-01358-f002:**
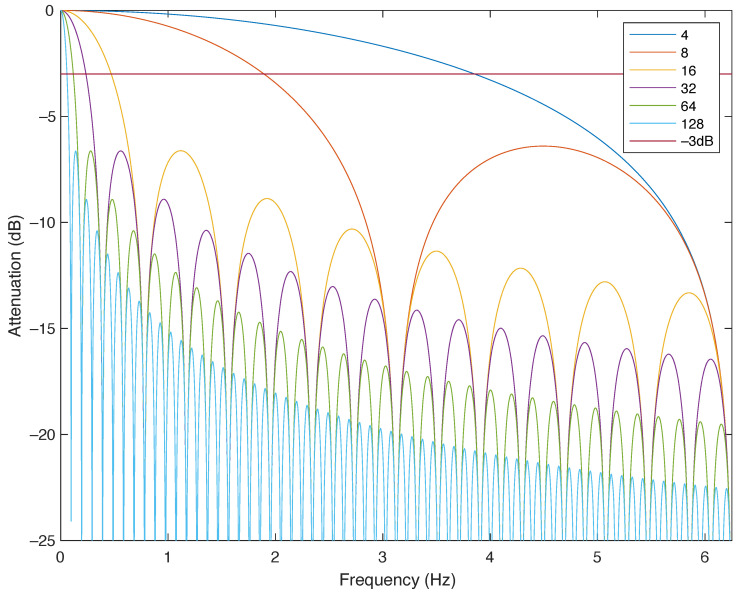
Low-pass FIR filter response based on an average of different lengths with a sampling rate of 12.5 Hz.

**Figure 3 sensors-24-01358-f003:**
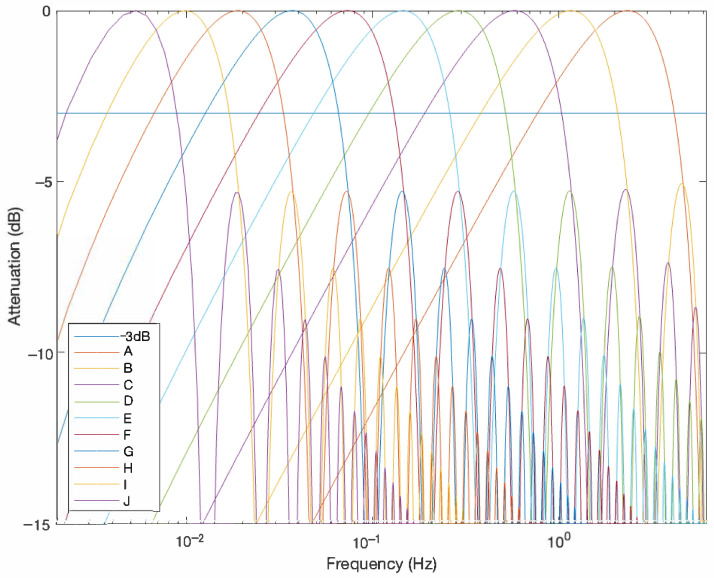
FIR filter bank response based on averaging and adjacent subtraction for different lengths with a sampling rate of 12.5 Hz.

**Figure 4 sensors-24-01358-f004:**
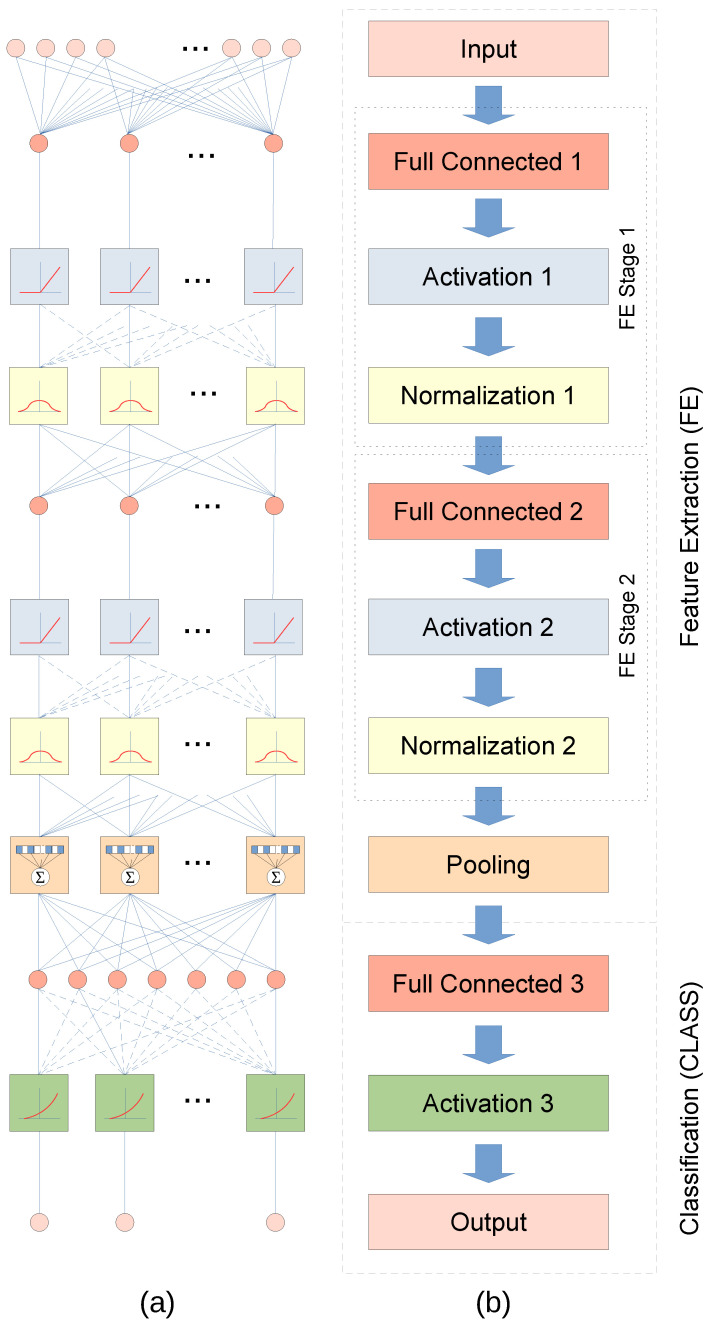
Proposed DNN scheme for filter selection in signal feature extraction schema: (**a**) diagram of primitives and connections, and (**b**) description of stages and layers.

**Figure 5 sensors-24-01358-f005:**
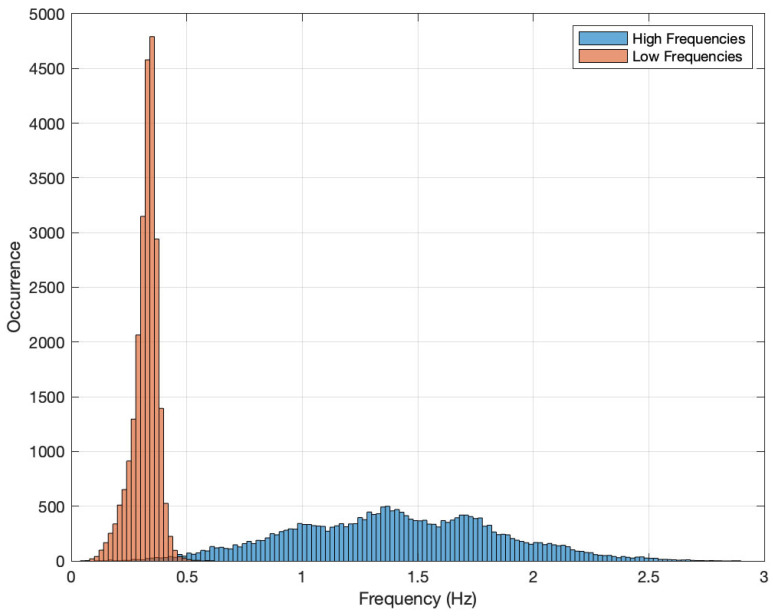
Histogram of frequencies measured during a day in periods of 2.56 s with a resolution of 0.1 Hz.

**Figure 6 sensors-24-01358-f006:**
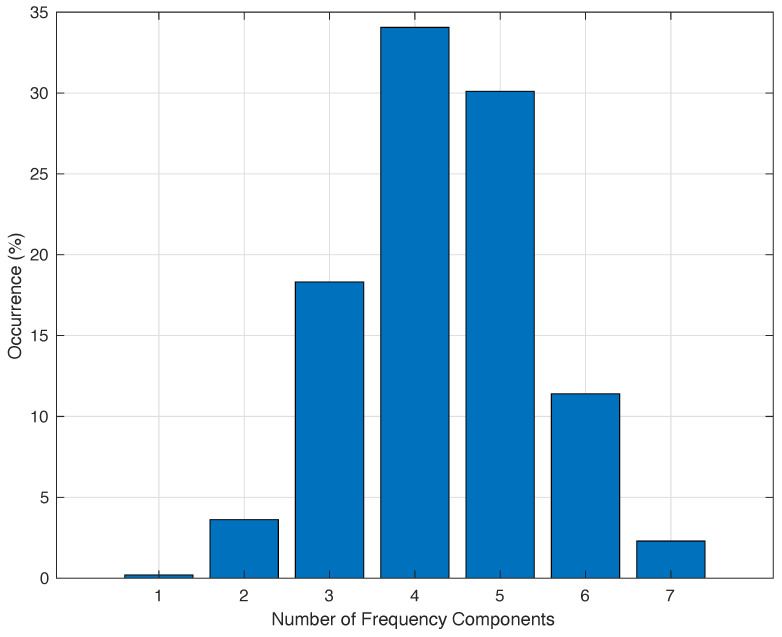
Distribution of the number of frequency components measured during a day in periods of 2.56 s with a resolution of 0.1 Hz.

**Figure 7 sensors-24-01358-f007:**
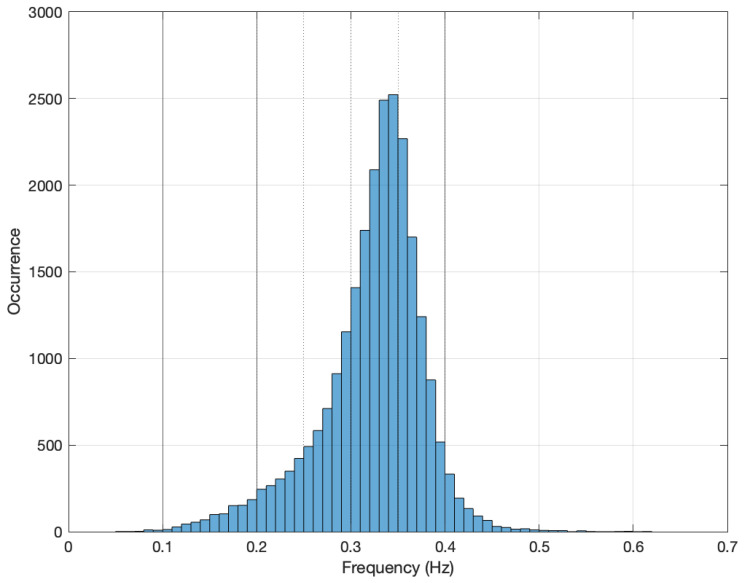
Fundamental frequency probability distribution, in terms of occurrence, for a field campaign measurement over a 24 h period (10 mHz bin resolution).

**Figure 8 sensors-24-01358-f008:**
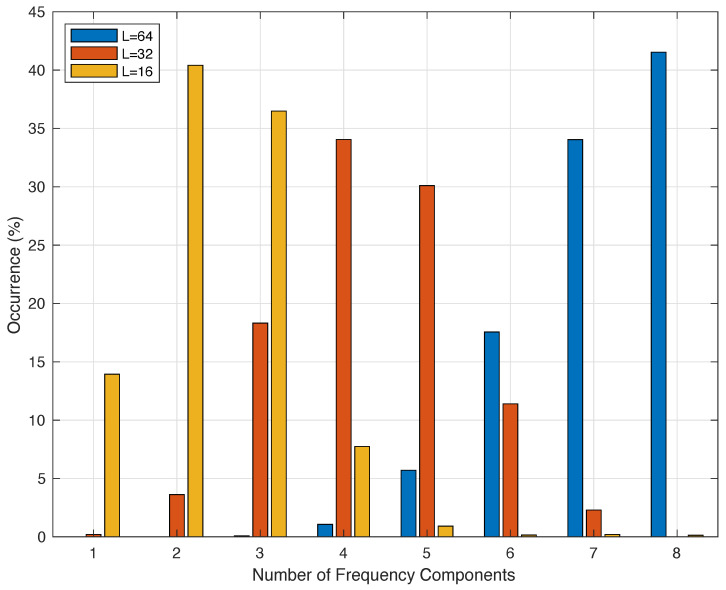
Distribution of the number of frequency components measured during a day in periods of 2.56 s for three different sample lengths.

**Figure 9 sensors-24-01358-f009:**
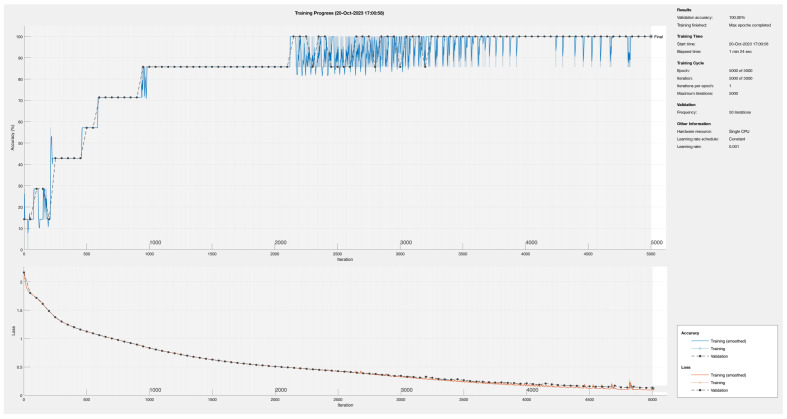
Training process for the proposed DNN using 16 neurons in fully connected Layers 1 and 2.

**Figure 10 sensors-24-01358-f010:**
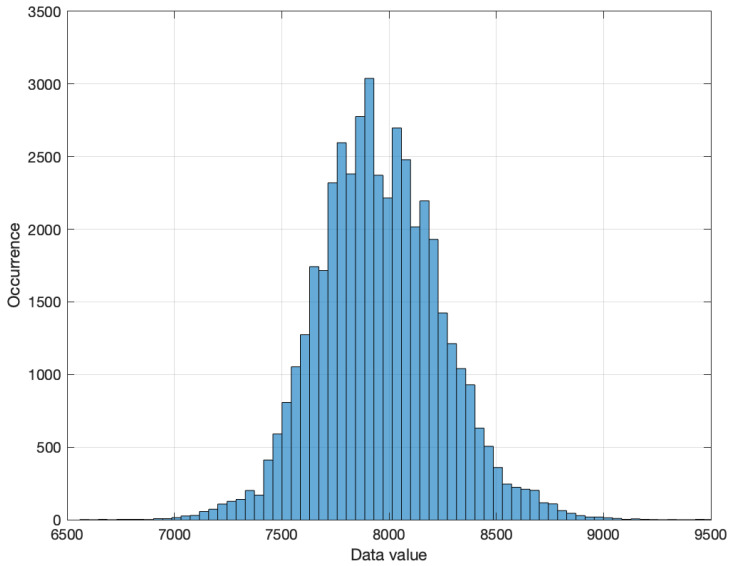
Input raw data occurrence distribution.

**Figure 11 sensors-24-01358-f011:**
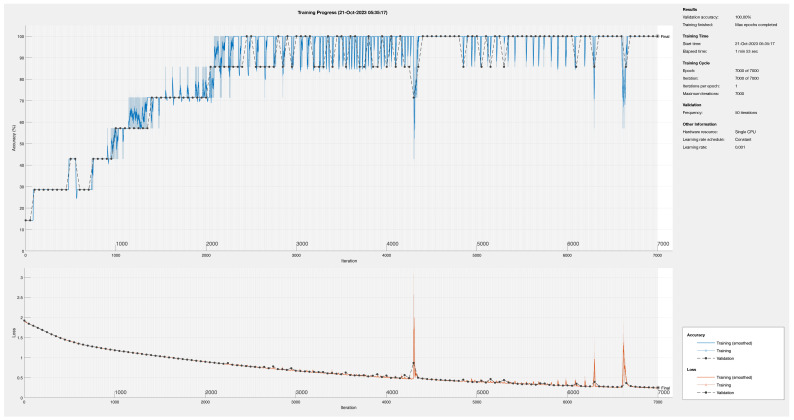
Training process of proposed ANN using only 3 neurons in Fully Connected 2 layer.

**Figure 12 sensors-24-01358-f012:**
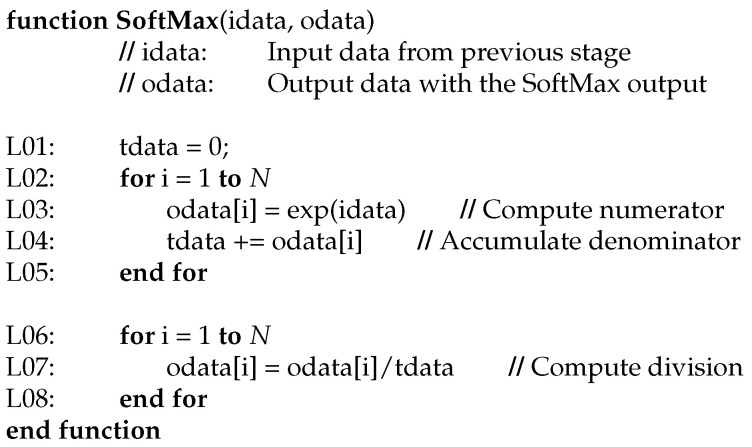
Pseudocode to implement SoftMax function.

**Table 1 sensors-24-01358-t001:** Published signal parameter extraction algorithms.

Reference	Year	Functions	Language	Domain	Input Filter	Equipment
[15]	2012	FIR, sin, sqrt	-	R	Fixed	Desktop PC
[16]	2017	GA	C	R	Fixed	Desktop PC
[12]	2018	Hilbert	Matlab	R	Fixed	Desktop PC
[11]	2019	Wavelet, log	Matlab	R	Filter Bank	Desktop PC
[17]	2020	GA	-	R	Fixed	Desktop PC
[13]	2020	Kal	Matlab	R	Fixed	FPGA, Cortex-M4 *
[20]	2021	FFT, GA, sqrt	C	R	Fixed	FPGA, Desktop PC
[21]	2021	Hilbert	Matlab	R	Filter Bank	Desktop PC
[10]	2022	FFT	Matlab	C	Fixed	Desktop PC
[22]	2022	Hilbert	Matlab	R	Filter Bank	Desktop PC
[9]	2023	FFT, IFFT, sin, sqrt	Matlab	C	Fixed	Desktop PC
[14]	2023	Kal, sqrt, covariance	Matlab	R	Fixed	Desktop PC
[6]	2023	FIR	C	Z	Fixed	Cortex-M0

IFFT: Inverse FFT; * DSP: TMS320C6711 (FPU); sin: trigonometric function; sqrt: square root function; Kal: Kalman filter; log: Logarithm

**Table 2 sensors-24-01358-t002:** Characteristics of the implemented filter bank using average filtering.

Filter			−3 dB		−0.58 dB	
ID	Length	Period	$F_{H}$	$F_{L}$	$F_{H}$	$F_{L}$
	(Samples)	(s)	(Hz)	(Hz)	(Hz)	(Hz)
A	4	0.64	4.191	0.782	3.178	1.573
B	8	1.28	2.083	0.391	1.573	0.788
C	16	2.56	1.044	0.189	0.788	0.394
D	32	5.12	0.521	0.099	0.394	0.197
E	64	10.24	0.262	0.048	0.197	0.098
F	128	20.48	0.130	0.024	0.098	0.049
G	256	40.96	0.066	0.012	0.049	0.025
H	512	81.92	0.033	0.007	0.025	0.013
I	1024	163.84	0.017	0.004	0.013	0.007
J	2048	327.68	0.008	0.002	0.007	0.004

**Table 3 sensors-24-01358-t003:** Number of machine cycles requires running advanced mathematical functions using a software implementation and compiled with gcc using the -O1 flag.

Math	Double			Float			Int32			Int16		
Function	Min	Avg	Max	Min	Avg	Max	Min	Avg	Max	Min	Avg	Max
$e^{x}$	7241	7400	7555	7335	7494	7649	7399	7556	7713	7398	7555	7712
$\sqrt{x}$	6673	6873	7242	6770	6951	7339	6800	6980	7362	6796	6976	7358
x/y	1373	1427	1485	281	282	293	127	136	152	125	134	150

**Table 4 sensors-24-01358-t004:** Number of machine cycles required executing common advanced math functions using an Arm Cortex-M0+ with the compiler optimization flag level 2 active.

Math	Double			Float			Int32			Int16		
Function	Min	Avg	Max	Min	Avg	Max	Min	Avg	Max	Min	Avg	Max
$e^{x}$	5282	5394	5282	5371	5482	5598	5404	5515	5633	5412	5525	5636
$\sqrt{x}$	4612	4732	5014	5092	4810	4689	4711	4830	4711	4710	4832	4711
x/y	903	937	903	196	197	202	84	89	99	84	89	99

**Table 5 sensors-24-01358-t005:** Proposed DNN approach computational effort evaluation.

Layer	Elements	Integer-Double		Integer-Float	
Name	(Neurons)	Cycles *	Operation	Cycles *	Operation
Input	32	0	Integer	0	Integer
Full Connected 1	5	966	Integer	966	Integer
Activation 1	5	70	Integer	70	Integer
Normalization 1	5	2883	Double	1232	Float
Full Connected 2	5	3276	Double	1942	Float
Activation 2	5	418	Double	331	Float
Normalization 2	5	2884	Double	1232	Float
Pooling	7	0	Double	0	Float
Full Connected 3	7	4008	Double	4008	Float
Activation 3	7	61,792	Double	53,920	Float
Outputs	7	0	Integer	0	Integer
Total Cycles		76,297		63,701	
Time @ 8 MHz		$9.54\times 10^{-3}$		$7.96\times 10^{-3}$	

* Included math operations and loop.

**Table 6 sensors-24-01358-t006:** Proposed ANN approach computational effort evaluation.

Layer	Elements	Integer-Double		Integer-Float	
Name	(Neurons)	Cycles *	Operation	Cycles *	Operation
Input	32	0	Integer	0	Integer
Full Connected 2	3	562	Integer	562	Integer
Activation 2	3	24	Integer	24	Integer
Normalization 2	3	1725	Double	736	Float
Pooling	7	0	Double	0	Float
Full Connected 3	7	4008	Double	4008	Float
Activation 3	7	61,792	Double	53,920	Float
Output	7	0	Integer	0	Integer
Total Cycles		68,111		59,250	
Time @ 8 MHz		$8.51\times 10^{-3}$		$7.41\times 10^{-3}$	

* Included math operations and loop.

## Data Availability

No new data were created or analyzed in this study. Data sharing is not applicable to this article.

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
