# Peer review of "An Edge Computing Application of Fundamental Frequency Extraction for Ocean Currents and Waves"

_sensors, 2024, doi:10.3390/s24051358_

Round 1

Reviewer 1 Report

Comments and Suggestions for Authors

The Manuscript entitled "An Edge Computing Application of Fundamental Frequency Extraction for Ocean Currents and Waves," delves into the development and enhancement of an artificial intelligence-based algorithm designed to augment the precision of ocean water current waves. The primary focus lies in attaining the fundamental frequency of ocean waves and currents, with a specific emphasis on addressing power consumption issues through the utilization of ultra-low power microcontrollers.

One notable strength of the paper is its acknowledgment of the inherent challenges in underwater applications, particularly the power constraints that necessitate the use of ultra-low power microcontrollers. The focus on optimizing energy efficiency in this context is creditable, considering the vital role power consumption plays in the feasibility and longevity of oceanic measurement devices.

However, the paper falls short in providing a comprehensive literature review, limiting the context within which the proposed algorithm operates. A more thorough examination of existing methods and their respective strengths and weaknesses would contribute significantly to the paper's credibility and situational relevance.

The utilization of a deep neural network within the realm of edge computing to ascertain the narrow bandwidth for filtering the fundamental frequency of ocean waves and currents is an interesting approach. However, the paper lacks clarity in detailing the specifics of the neural network architecture and the reasoning behind choosing this particular methodology. Without a clear exposition of the neural network's design and performance metrics, readers are left questioning the reliability and robustness of the proposed solution.

Moreover, the manuscript mentions the assumption that existing extraction algorithms presume a fixed bandwidth for the processed signal. Yet, the paper does not adequately establish why this assumption is problematic or how the proposed deep neural network addresses this limitation. A more in-depth discussion on the inadequacies of conventional methods and the novel contributions of the proposed algorithm would significantly strengthen the paper's scientific merit.

Reviewer 2 Report

Comments and Suggestions for Authors

These are the comment need to be addressed.

1. The abstract needs to be rewritten with the results

2. What is the role of GNN in defining a graph-based architecture in the study?

3. Novelty missing in the manuscript.

4. Implications of the study need to be developed 

Comments on the Quality of English Language

its ok 

Reviewer 3 Report

Comments and Suggestions for Authors

The paper tackles a significant issue, yet there are certain aspects that require further examination.

The abstract is somewhat vague in its description of the proposed algorithm and its performance improvement. It would be more informative to provide concrete details about the algorithm's architecture, the training data, and the metrics used to evaluate its performance.

 The introduction fails to adequately emphasize the novelty and significance of the proposed AI approach. It should provide a clearer and more focused explanation of how the AI approach differs from existing methods and the benefits it offers. It doesn't provide any references that could validate the problem.

The paper fails to provide a clear and convincing argument for the superiority of the proposed neural network approach compared to existing methods. It is unclear how the neural network addresses the limitations of other approaches and what specific advantages it offers.

The paper describes the implementation of the neural network without clearly connecting it to the specific problem it aims to solve. Moreover, the evaluation criteria used to assess the neural network's performance are not well-defined, and the results of the experiments are not discussed in detail.

Comments on the Quality of English Language

n/a

Reviewer 4 Report

Comments and Suggestions for Authors

The main question addressed by the research is how to enhance the accuracy of ocean water current meters, which traditionally face limitations in power consumption due to the use of ultra-low power microcontrollers. The focus is on developing a smart algorithm based on artificial intelligence, particularly a deep neural network, to determine a narrow bandwidth for filtering the fundamental frequency of ocean waves and currents in an edge computing context.

The topic is both original and highly relevant in the field. It addresses a critical issue in oceanography - the accurate measurement of ocean currents and waves, which has applications in various domains including environmental monitoring, navigation, and disaster management. The use of edge computing and a deep neural network to optimize power consumption and improve accuracy represents a novel approach that contributes to the evolving landscape of oceanographic instrumentation.

The research adds significant value to the subject area by introducing a novel methodology for fundamental frequency extraction in underwater applications. The integration of edge computing and deep learning distinguishes this research from other published materials. By leveraging advanced technologies, the paper presents a practical solution to the power consumption challenge, potentially advancing the capabilities of oceanographic instruments for more accurate data collection.

There are a few suggestions that could help improve the content and clarity of the research:

  1. The conclusions drawn in the paper are generally consistent with the evidence and arguments presented. However, providing more details on the performance metrics and comparing the proposed approach with existing methods would strengthen the conclusions. It would also be beneficial to discuss any potential limitations or challenges faced during the implementation of the algorithm.

  1. The references seem appropriate and cover a range of relevant literature in the fields of oceanography, edge computing, and artificial intelligence. However, a more in-depth discussion comparing the proposed approach with existing methods would enhance the scholarly aspect of the paper.

Round 2

Reviewer 2 Report

Comments and Suggestions for Authors

All the comments were well addressed, and now the manuscript has moved to the next step.

Author Response

Thanks for your comments.

We also have updated the paper following the recommendations from Reviewer 3 and also the academic editor.

Reviewer 3 Report

Comments and Suggestions for Authors

The authors have made progress in refining the manuscript, but there is still room to enhance its quality.

Introduction

The introduction to the manuscript needs to be strengthened in several areas.

  • Citations: The introduction makes numerous claims without providing any references to support them. This makes the manuscript less credible and weakens the overall argument. It is important to cite relevant previous work to establish the context of the current research and provide evidence for the author's claims.
  • Contribution: The introduction should explicitly state the contribution of the author's work. This should involve clearly outlining the problem addressed by the research and how it differs from or improves upon existing approaches. Without a clear statement of contribution, it is difficult for readers to understand the purpose of the manuscript and its significance.

Related Work

The related work section needs to be reorganized and improved in several ways:

  • Contextualization: Provide a paragraph for introducing the section. The section 2. starts with 2.1
  • Separation: Clearly distinguish between related work and background concepts. Related work should focus on previous studies that are directly relevant to the current research, while background concepts should provide essential information for understanding the problem and the proposed approach.
  • Research Gap: Clearly identify the research gap that the current work fills. This should be done in the last paragraph of the related work section. By explicitly stating the gap, the authors can emphasize the significance of their contribution and the originality of their approach.

Author Response

Thanks for your review. Please see the attachement.
